# Local Therapy and Reconstruction in Penile Cancer: A Review

**DOI:** 10.3390/cancers16152704

**Published:** 2024-07-30

**Authors:** David Zekan, Rebecca Praetzel, Adam Luchey, Ali Hajiran

**Affiliations:** 1School of Medicine, West Virginia University, Morgantown, WV 26506, USA; adam.luchey@hsc.wvu.edu (A.L.); ahajiran@hsc.wvu.edu (A.H.); 2College of Osteopathic Medicine, Liberty University, Lynchburg, VA 24502, USA; rpraetzel@liberty.edu

**Keywords:** penile cancer, partial penectomy, radical penectomy, Mohs surgery, topical therapy, glans resurfacing, glansectomy, penile reconstruction

## Abstract

**Simple Summary:**

Squamous cell carcinoma of the penis is a rare yet distressing condition representing less than 1% of cancer diagnoses in the United States annually. It is largely associated with human papillomavirus infection, lack of circumcision, and poor hygiene, among other factors. When detected early, local therapies for penile cancer offer robust response and cure rates, but they can be disfiguring, leading to psychologic, social, and functional distress. Herein, we explore local therapies, including topical drugs, laser therapy, and excisional procedures, and more radical surgeries for squamous cell carcinoma of the penis as well as advanced reconstructive techniques, including skin grafting and the creation of a new penis to maintain functional status and cosmesis.

**Abstract:**

Local therapy for penile cancer provides robust survival and can preserve the penis functionally and cosmetically. Interventions must target the appropriate clinical stage. We reviewed studies regarding the primary therapy in penile cancer, from topical therapy to radical penectomy, and reconstructive techniques. Topical therapy (5-FU or Imiquimod) provides a robust oncologic response in patients with Ta or Tis disease. Multiple laser therapies are available for localized patients and those with low-grade T1 disease. There is a non-trivial risk of progression and nodal metastases in poorly selected patients. Wide local excision provides an oncologically sound option in patient with up to T1 disease; less evidence exists for Mohs microsurgery in the setting of penile cancer. Increasingly aggressive approaches include glansectomy and partial/radical penectomy, which provide 5- and 10-year cancer-specific survival rates of over 80%. Meticulous reconstruction is necessary for the durable function of the remaining penis. Preservation of voiding and sexual function occurs via penile skin grafting, glans resurfacing, creation of a functional penile stump, and phalloplasty with a penile implant. Perineal urethrostomy provides an alternative in pathology demanding extensive partial or radical penectomy, and a durable option for seated voiding. Clinical suspicion and timely diagnosis are paramount in terms of management as less-invasive options for earlier-stage disease develop.

## 1. Introduction

An overwhelming majority of penile masses are histologically squamous cell carcinoma (SCC). An estimated 2050 new cases of penile cancer were diagnosed in 2023, with 470 deaths [1]. Overall, the incidence is increasing in the United States, particularly in the South and areas of low socioeconomic status [2]. It occurs most commonly in men in their 60–70s [2]. The risk factors associated with the development of SCC of the penis include smoking, human papilloma virus (HPV) infection, phimosis, lack of circumcision, and poor hygiene [3,4,5]. It is characterized by early metastatic spread to the lymph nodes, and survival varies widely depending on the stage at diagnosis and nodal involvement [6]. With the increasing incidence as above, early detection and optimization of local therapy are crucial, particularly considering the psychological impact of the loss of upright voiding and the sexual dysfunction associated with radical surgery [7]. In fact, some men who have undergone therapy for penile SCC retrospectively prioritize maintenance of potency over long-term survival as the effects of penectomy on quality of life are significant [8]. The aggressiveness of the therapy correlates conversely with the quality of life with regard to overall health status, sexual health, and physical functioning [9].

The use of local therapy as opposed to radical therapy is largely dictated by pathology on biopsy. Specifically, patients with carcinoma in situ (Tis) and non-invasive disease (Ta) are appropriate candidates for topical therapy, wide local excision, laser therapy, complete glansectomy, or Mohs surgery in select cases. In patients with T1 disease on biopsy, candidacy for the above is dictated by the histologic grade, with grade 1–2 being amenable to those and radiation. However, T1 grade 3–4 requires more radical therapy in the form of at least wide local excision but more commonly partial or radical penectomy. T2 or greater disease on biopsy requires the most radical therapy, which is dictated by the clinical nodal status and often includes radical penectomy with perineal urethrostomy (PU), sometimes preceded by neoadjuvant platinum-based chemotherapy [10]. Naturally, local therapy is dictated by the location of the penile lesion. The most common site for penile SCC is the glans (34.5%), followed by the prepuce (13.2%), penile shaft (5.3%) and overlapping sites (4.5%), while 42.5% of cases occur at non-specific sites [2,8]. Moreover, 62% of disease diagnosed in the United States is localized to the penis, demonstrating the importance of organ-sparing principles [2,8].

Organ-sparing principles are largely divided into non-surgical (radiation, topical immune-response modifiers) and surgical management, with the latter requiring reconstruction for practical voiding, acceptable cosmesis, and, in many cases, potency. Given the undesired consequences associated with non-surgical management, including urethral stricture from high-dose radiation and difficulties in surveillance/detection of recurrence following the use of topical immune modulators, as well as tailoring of reconstructive techniques, the popularity of conservative surgical management is increasing [8,11]. Herein, we review the efficacy of local therapies, compare their oncologic outcomes to radical penectomy, and provide an overview of pertinent reconstructive techniques.

## 2. Materials and Methods

This review evaluates local therapy and reconstructive techniques for patients with localized penile SCC. The authors performed a review across electronic databases using PubMed, EMBASE, and the Cochrane Database of Systematic Reviews. Our search utilized terms pertinent to localized penile cancer and penile reconstruction. Specifically, we subdivided local (primary) therapy into categories including topical therapy, laser therapy, wide local excision, glansectomy, Mohs surgery, and partial/radical penectomy. Reconstructive techniques were divided into categories including penile skin grafting, glans resurfacing and creation of neo-glans, functional penile stump, perineal urethrostomy, and phalloplasty and penile implant. We also reviewed the reference lists of eligible studies and recent review articles to ensure no pertinent literature was overlooked. We excluded works published prior to 1997, aside from early descriptions of Mohs micrographic surgery published in 1985 due to the limited nature of the literature in this space (Figure 1).

## 3. Results

### 3.1. Primary Therapy

#### 3.1.1. Topical Therapy

Topical therapy is reserved for patients with Tis or Ta SCC of the penis on biopsy. Generally, this takes the form of Imiquimod 5% applied at night three times per week for 4–16 weeks or 5-fluorouracil (5-FU) cream 5% applied twice daily for 2–6 weeks [10]. Such therapy provides an upfront minimally invasive option for patients who prioritize penile-sparing therapy. For biopsy-proven Cis specifically, Hajiran et al. reviewed twenty patients treated over a ten-year period at a tertiary cancer center. Most patients (n = 17; 85%) received topical 5-FU with a median follow-up of 18 months. Complete response (CR) was achieved in 65%, partial response in 25%, and no response in 10%. Half of the patients had either incomplete response or relapse requiring use of other therapies. The median recurrence-free survival was 14 months. Nonadherence to therapy was common in the study and had a decreased CR rate (28.6%) [12]. In a 2022 Swedish study, both the incidence of Cis overall and the use of topical therapy in the form of both 5-FU and Imiquimod increased over time. A total of 1113 cases were reviewed from 2000 to 2019, with an increase in incidence from 1.40 to 2.37/100,000 during the study period. Local therapy was more common than destructive therapy in the last five years of the study period when compared to the first five [13]. No randomized clinical trials exist comparing the response rates and durability of 5-FU vs. Imiquimod. Initial studies showed durable responses to local 5-FU therapy, with normal post-treatment biopsies in the originally studied groups of three patients and seven patients at 20–60 months and 70 months, respectively. In a larger retrospective study of 86 patients followed for ten years, 5-FU had a 50% complete response compared to 44% of patients treated with Imiquimod. Moreover, 31% of patients treated with 5-FU had a partial response, while Imiquimod had a 56% non-response [14]. Both 5-FU and Imiquimod are relatively well-tolerated, with adverse effects including hypersensitivity, irritation, and pain, and itching, erythema, hypopigmentation, tenderness, bleeding, and crusting [15]. Due to the difference in response rates for the respective therapies, Imiquimod is often reserved for partial responders or patients with recurrence after 5-FU therapy [15]. Combination therapy has also been explored, specifically topical Imiquimod therapy followed by CO_2_ laser ablation in a small series (n = 10) over a five year period, with no residual tumor in six patients, stable disease in two patients, and progressive disease in two patients over a mean follow-up of 26 months. Interestingly, all the complete responders had HPV-related lesions [16].

#### 3.1.2. Laser Therapy

Although also plagued by a lack of large, randomized controlled studies, laser therapy for SCC of the penis is recommended in patients with clinical stage Tis, Ta, and T1 grades 1–2 penile cancer. The guidelines underscore the use of perioperative acetic acid (3–5%) to identify areas of HPV-infected skin as targets for treatment. Similarly, use of a smoke evacuator is emphasized to reduce exposure to HPV and other viral particles [10]. A host of lasers can be utilized with low complication rates and cosmetic results comparable to other penile-sparing modalities, as well as better sexual outcomes than other local therapy. The laser types include neodymium: yttrium-aluminum-garnet (Nd:YAG), thulium: yttrium-aluminum-garnet (Tm:YAG), and CO_2_ lasers. The largest case series of Tis/T1 SCC of the penis managed with a CO_2_ laser included 224 patients with a ten-year recurrence rate of only 17.5%, with amputation rate of 5.5% and acceptable function results [17]. The limitations of laser therapy are apparent in aggressive disease. Tang et al. retrospectively reviewed 161 patients treated with either a CO_2_ or Nd:YAG laser for up to cT2 SCC of the penis. They show a five-year recurrence-free survival of 46%, with a risk of nodal recurrence varying from 2% in Tis to 22% in T2, underscoring the importance of clinical nodal staging and use of laser therapy in only appropriately selected patients [18]. Less data exist regarding the use of Tm:YAG lasers in penile cancer, but this shorter wavelength laser (2000 nm) with a short penetration depth and rapid vaporization of tissue provides another option for the treatment of superficial penile tumors [19]. Musi et al. reported the functional and oncologic outcomes of patients with pTis-pT3 disease on final pathologic specimens treated with Tm:YAG lasers. At a median follow-up of 24 months, four (17.4%) patients had a recurrence, of whom three (13.0%) and one (4.3%) patient developed an invasive or in situ recurrence, respectively. After treatment, six (26.1%) patients reported conserved penile sensitivity, while thirteen (56.5%) and four (17.4%) patients experienced a better or worse sensitivity after ablation, respectively. Moreover, 82.6% of the patients reported sexual activity within the first month after treatment [20]. Perhaps the longest follow-up aside from the above is provided by Meijer et al. and Windahl and Andersson. The former followed 44 patients from 1986–2003 with biopsy-proven SCC of the penis; the mean follow-up was 52.1 months, and on biopsy, 21 had T1 disease, 17 T2, and 6 Tis, respectively. Local recurrence (in the treated area) occurred in 48% of patients, and 20% had recurrence elsewhere on the glans penis, which was treated with either repeat laser or partial amputation. Nodal metastases were detected in 10 patients, 8 of whom had T2 disease on biopsy. The authors suggest the use of the clinical stage, rather than the grade, as an important prognostic predictor of nodal disease [21]. Windahl and colleagues performed similar prospective surveillance of 67 men from 1986–2002, for a median of 42 months, treated with a combination CO_2_ and Nd:YAG laser. In their study population, 21 patients had pTis, 2 pTa, 23 pT1, 19 pT1, and 22 pT3. During the follow-up, 59 patients were alive and 8 had died of penile carcinoma (2) and concurrent disease (6). Of the thirteen patients (19%) with local recurrence during the study period, ten underwent repeat laser treatment successfully [22]. The same group was interviewed regarding sexual function and satisfaction from six months to fifteen years post-operatively (median 3 years). Of 40 patients (87%) who had been sexually active before treatment, 75% had resumed activities at the time of the interview. Only 22% of patients reported decreased erectile function. In addition, 50% of patients were satisfied/very satisfied with their sexual life. Dyspareunia was reported in 10% [23]. In appropriately selected and surveilled patients, laser therapy provides a satisfactory oncologic, cosmetic, and sexual outcome for SCC of the penis.

#### 3.1.3. Wide Local Excision

Wide local excision (WLE) for SCC of the penis is well suited for early-stage penile cancer confined to the skin with little to no invasion (cTis, Ta, or T1). The surgical margins are a hotly debated topic, but the guidelines state that these depend on the location of the tumor, emphasizing that circumcision alone may be sufficient for tumors of the distal prepuce [10]. Often, wide excision requires grafting, as outlined below. The classical teaching was that a 2 cm gross margin was necessary when performing WLE to ensure complete resection. Anderson et al. reviewed 21 cases of penile SCC treated by a single surgeon from 2010 to 2019. The pathologic stage was Tis in 10.5% (n = 2) of patients, T1a in 26.3% (n = 5), T1b in 15.8% (n = 3), T2 in 36.8% (n = 7), and T3 in 10.5% (n = 2), respectively. The histopathologic tumor grade was Cis only in 9.5% (n = 2) of patients, grade 1 in 14.3% (n = 3), grade 2 in 38.1% (n = 8), and grade 3 in 38.1% (n = 8). An algorithm was developed for the margins based on the grade at biopsy, with macroscopic margins of 5 mm for grade 1, 10 mm for grade 2, and 20 mm for grade 3, respectively. During the study period, only a single patient required completion radical penectomy for oncologic control due to pT3 grade 1 disease on pathology following initial resection. With a median margin of 7 mm, the overall survival, cancer-specific survival, metastasis-free survival, and local recurrence-free survival were 94.6%, 94.6%, 81.0%, and 92.3%, respectively, demonstrating excellent oncologic control [24]. In terms of sexual function, WLE is superior to glansectomy when utilizing the International Index of Erectile Function (IIEF), with no significant reduction in the IIEF following WLE in a series of 41 patients [25]. Cilio et al. corroborate these results in 34 patients, with patients who underwent WLE versus glansectomy demonstrating a higher rate of erectile dysfunction and sexual impairment [26]. Utilization of WLE in cTis penile cancer is widely debated, and such pathologic findings are considered a contraindication for WLE by some. Other relative contraindications include disease occupying over half of the glans, close proximity to the urethral meatus, and urethral involvement [27]. Overall, for pT1 and T2 tumors, local recurrence is relatively common, but the five-year recurrence-free survival is 63% [27]. Thus, with appropriate surveillance, WLE provides an effective method for management of at least up to cT1 SCC of the penis.

#### 3.1.4. Glansectomy

Glansectomy (or glans resurfacing) is the preferred therapy in only a select subset of patients with SCC tumors of the glans or distal prepuce, but it remains an important consideration because penile cancer occurs distally in up to 80% of cases [27]. For patients with clinical Ta or Tis disease, glansectomy is considered a category 2B recommendation (based upon lower-level evidence, but with NCCN consensus that intervention is appropriate). For clinical T1, grade 1–2 disease, it is not recommended unless required to ensure complete tumor eradication with adequate negative margins, as determined on frozen section from the cavernosal bed and urethral stump [10]. Glans resurfacing (resection of the glans epithelium and subepithelium until the margins are macroscopically clear with subsequent grafting) is an attractive option for Tis disease, with 28% of patients requiring further intervention for positive margins in one series but without compromising oncologic outcomes [15]. As discussed at length below, glansectomy is often accompanied by either partial- or full-thickness skin grafting to create a neo-glans. Baumgarten et al. highlight a large case series and international cohort of 1188 patients with SCC of the penis treated with penile-sparing surgery. Herein, they highlight a discrepancy between European and American guidelines, in which European guidelines list complete glansectomy as appropriate therapy in patients with pT2 disease confined to the glans. Of the 282 pT2 patients in their cohort, 60.9% were treated with glansectomy, with 85.7% of the pT2 patients experiencing recurrence within 5 years [28]. Several smaller European cohort studies with shorter follow-up exist, one in which 72 consecutive patients underwent glansectomy for penile carcinoma. At a mean follow-up of 27 months, there were three recurrences at 4, 19 and 28 months, respectively. In this cohort, 51% of the men had T2 disease [29]. Most studies boast a disease-specific survival of >90% during limited follow-up. Of patients with pT1-2 disease, the five-year recurrence-free rates range from 78.0 to 95.8% [30]. This large variation in recurrence rates amongst studies again highlights the importance of patient selection and negative surgical margins.

#### 3.1.5. Mohs Surgery

During Mohs micrographic surgery (MMS), the tumor is excised horizontally with multiple frozen sections and microscopic evaluation until the bed is free of disease. This allows for clear margins and the preservation of uninvolved penile tissue for good cosmetic and functional results [27]. It is considered a category 2B recommendation for SCC of the penis and an alternative to WLE in select cases. It allows for increased precision in the detection of negative resection margins, but the success rates decline with an increased stage of disease. However, this does provide an alternative for proximal shaft lesions that are low risk on biopsy and would otherwise require radical penectomy based solely on location. The recurrence rates are higher than other cutaneous malignancies but comparable to other organ-sparing techniques [10,31]. Initial reports on penile MMS came about in 1985. The results were promising, with distal (glans or prepuce) lesion five-year cure rates of 81%; however, this fell to 57% with shaft lesions. The lesion size and prior treatment also significantly predicted the outcomes. The cosmetic and functional results were favorable [15,32]. Alcala et al. corroborate the above in a 2010 to 2020 review of 43 patients with Ta-T2 disease (including Tis) undergoing MMS, none of whom had positive margins following resection. The stage distribution was Ta in 4.7%, Tis in 58.1%, T1a in 14.0%, T1b in 7.0%, and T2 in 16.3%. The overall local recurrence rate was 2% (n = 1) at a median of 47 months. Local recurrence occurred in no patients with Ta-T1 disease at 5 years. The local recurrence rates for T2 patients were 14% at one year [33]. Overall, the five-year recurrence-free rates in men with penile cancer treated with MMS are estimated to be between 80 and 90% [30]. This again underscores the importance of appropriate patient selection when performing penile-sparing surgery for pathologically aggressive tumors.

#### 3.1.6. Radiation Therapy

Radiation has a limited role in local therapy for penile cancer, but it is an option for invasive penile SCC (≥T1, grade 1–2). Both EBRT and brachytherapy can be utilized in the post-circumcision setting for T1–2, N0 disease. Concurrent chemotherapy is often administered in this setting and should universally be used in T3–4 or N+ disease. Post-surgical EBRT is also an option with a positive primary site margin following penectomy [10]. EBRT delivers radiation doses of 60–75 Gy, with 5-year local control rates of 40–70%. CSS in reported series range from 58 to 86% [15]. Brachytherapy can be used to deliver radiation doses from 50 to 65 Gy. The 5-year local control rates in the reported series range from 70 to 87% and there is a 5-year CSS of 72–88%. Well over half of patients are able to maintain their penises during the follow-up periods [15]. Important considerations in radiation patients are the above-described circumcision prior to therapy to avoid phimosis as well as late sequelae, including meatal stenosis and ulceration of the penile skin and soft tissue [15].

#### 3.1.7. Partial/Radical Penectomy

The widely accepted gold standard for invasive penile SCC (>T1, grade 1–2) is partial or radical penectomy. Partial penectomy may also be considered in patients with T1, grade 1–2 disease [10]. These procedures represent the comparator for all the penile-sparing approaches, making it of the utmost importance to define their oncologic outcomes. Multiple direct comparisons of recurrence and survival exist. The original experience, which contributed substantially to the current guidelines, was performed at Heidelberg, in which invasive tumors treated from 1968 to 1994 were reviewed. In this series of 51 patients, those with T1 disease treated with partial or radical penectomy had no recurrences, compared to 56% in penis-preserved patients. For T2 tumors, the local recurrence rate was 100% (organ preservation) versus 20% (amputative procedures) [34]. More contemporary European data evaluating 203 patients from 2000 to 2011 with a median follow-up of 61 months suggest a similarly increased local recurrence. Management included penile preserving surgery (49%), partial penectomy (24%), radical penectomy (24%), and chemotherapy or radiotherapy for metastatic disease (3%). Most patients presented with disease amenable to penile preservation. However, after organ-preserving surgery, the local recurrence rate was 18% (compared with 4% for amputative surgery), with 94% of recurrences occurring within three years. The 5-year CSS was 85% and the 10-year CSS was 81% [35]. Perhaps the largest study assessing the surgical modality and recurrence/survival included 859 patients with invasive penile SCC (T1–3) treated from 1956 to 2012. As one may expect, the incidence of penile amputations over time decreased. The 5-year local recurrence after penile preservation was 27%, while after (partial) penectomy it was 3.8%. Patients treated with penile preservation showed no significant difference in survival compared to patients treated with (partial) amputation after adjusting for the tumor stage. Importantly, in the penile preservation group, local recurrence was not associated with CSS [36]. These data are corroborated by a recent report from Lindner et al., who evaluated the surgical modality, recurrence, and survival in 55 patients treated at their institution with penile SCC and at least 24 months of follow-up (mean 63.7 months). Organ-sparing was performed in 26 patients (47.2%) and partial or total penectomy in 29 (52.8%). Patients in the penectomy group were significantly older, with a higher rate of advanced tumor stage (≥pT2: 44.8% vs. 11.5%), and the local recurrence rate was 42.3% (n = 11) following organ-sparing surgery compared to 10.3% (n = 3) after penectomy. No significant differences existed between the two groups regarding metastasis-free and overall survival [37]. These results suggest that the advent of organ preservation as opposed to extirpative therapy for SCC of the penis certainly has a role in management, perhaps even in T2 disease in accord with the European guidelines, but that post-operative surveillance is imperative in patients undergoing penile-sparing therapy (Table 1).

### 3.2. Reconstructive Techniques

#### 3.2.1. Penile Skin Grafting

Penile skin-grafting techniques have been successful for many patients and result in positive cosmetic outcomes. Both split-thickness skin grafts (STSGs) involving the epidermis and the superficial dermis, and full-thickness skin grafts (FTSGs) involving the entire epidermis and dermis, have shown promise in regards to appearance and function [38].

Split-thickness skin grafting is a reconstructive option for patients with penile cancer post-glansectomy. This method can be used for larger areas, since there is less metabolic demand to the graft, with a thinner layer of tissue being used [38]. The donor site is typically the medial thigh [39]. The procedure starts with prepping the skin with antiseptic, followed by sterile saline. The donor skin is removed with firm pressure of the dermatome, and the skin should be held taut with traction to maintain its integrity. Meshing is performed if it is a larger graft. The skin is placed on the graft site and secured with sutures or staples. The graft can then be bolstered and bandaged [38]. Oncologic control is preserved in patients undergoing this procedure for locally advanced penile cancer of the glans [40,41]. Urinary and sexual functions are also maintained or only minimally affected [40,42].

Full-thickness skin grafting of the penis is usually performed under general anesthesia [43]. The donor site is typically the inguinal region or the scrotum [39,43]. These donor sites will have more scarring than in STSGs due to the thickness of the graft. The graft is generally 10–20% larger than the recipient site when utilizing this method because of the higher incidence of primary contraction of the skin [38,39]. It is also important that the penis remains erect during the procedure to decrease the likelihood of wrinkling that could lead to graft rejection. Upon obtaining the full-thickness graft, subcutaneous fat is removed. Once hemostasis is obtained, the graft is sutured and bolstered with gauze to optimize the apposition and then covered with pressure dressing [38,43]. The results are safe and effective, improving erectile function, as indicated by the increased scores on the International Index of Erectile Function Questionnaire (IIEF), with minimal complications [43,44]. However, there is generally a higher incidence of graft failure in FTSGs compared to STSGs.

#### 3.2.2. Glans Resurfacing and Creation of Neo-Glans

Total glans resurfacing (TGS) is a promising reconstructive option for localized, superficial penile cancer (CIS, Ta, and T1 grade 1/2). The standard procedure is well described in detail by Pappas et al. and Palminteri et al. [45,46]. It begins with circumcision, followed by removal of the glans epithelium and underlying tissue to the level of the corpus spongiosum. The spongiosum is subsequently biopsied to ensure no invasion of this depth [45,46]. Complete removal of these layers has the oncologic benefits of decreased recurrence and the obtention of an undamaged sample available for pathology [47,48]. An STSG from the thigh is obtained and fenestrated, then wrapped over the recipient site and sutured into place. Appropriate gauze dressing and compression is applied, and a foley, and sometimes additionally a suprapubic catheter, is placed. Partial glans resurfacing would be indicated over TGS when less than half of the glans has been invaded, and the procedure is identical aside from the extent of the glans being removed [45].

In a 2017 prospective study of TGS by O’Kelly et al., zero of the nineteen patients undergoing TGS had postoperative complications, and while nearly all the patients experienced complete graft take, one had graft breakdown attributable to graft site dermatillomania [49]. In a 2020 retrospective study by Falcone et al. including twenty-six patients undergoing TGS, zero experienced intraoperative complications, and only a single patient experienced post-operative genital wound infection [48]. Sexual function was restored in each of the patients who reported being sexually active prior to the procedure, with an improved median IIEF score postoperatively as well [49]. Falcone et al. suggest further research on the preservation of sexual function with this procedure, but with a generally preserved anatomy and minimal postoperative complications, this minimally invasive surgery is favorable for penile cancer [48].

Construction of a neo-glans is indicated post-penectomy for invasive penile carcinoma [9] and post-glansectomy for disease limited to the glans [41,50]. The new glans is created via either STSG or acellular dermal matrix (ACM). An ACM, similar to an extracellular matrix, is prepared by removing a layer of epidermis from the donor site. A hypotonic solution is applied to lyse the cells, which are then decontaminated of antigens that could lead to possible immunologic rejection [51]. The donor site is typically the thigh [45,46,47] but can also include oral buccal mucosa, thigh, scrotal skin, and lower abdominal skin [50].

In 2018, Weibel et al. first suggested the use of a tunica vaginalis for a neo-glans [50]. Corpus cavernosum using a tunica vaginalis testis graft (TVTG) was chosen partially because their patient needed reconstruction of a substantial portion of the urethra. They describe the benefits of this allograft as being easy to harvest, being more cost-effective than some other grafts, possessing quicker procedural time, and resulting in more discrete scarring. However, in this case, a redo with an STSG was performed, and further study is needed to investigate the graft take [50].

Neo-glans reconstruction yields satisfactory results for patients. Palminteri et al. report that all of their twenty-one patients were pleased cosmetically, experienced restored sexual function with the caveat of some diminished sensitivity, and had no local recurrence at an average follow-up of forty-five months [46]. Yao et al. prospectively studied twenty-five patients who received neo-glans reconstruction post glansectomy for squamous cell carcinoma and found 92% had cosmetic satisfaction [41]. Postoperatively, two of the twenty-five experienced urethral meatus stricture and two had local recurrence at follow-up. Overall, positive cosmetic, voiding, and sexual function outcomes suggest this can be an ideal reconstructive modality for patients being treated for superficial penile cancers.

#### 3.2.3. Creation of Functional Penile Stump

Creation of a functional penile stump is an optional modification following partial penectomy. Following amputation of the necessary area, the “parachute” technique, illustrated by Korkes et al., can be implemented [52]. It involves ventral urethral spatulation and suturing the “V” skin flap from ventral to dorsal, so as to approximate the edges while preserving a urethral meatus [52]. Aside from the retention of most penile function, the benefit to using this reconstructive technique is its universality; while other procedures may be applicable for a smaller cohort of patients or require more niche surgical training, the parachute technique can be used for nearly any patient with indications [52,53]. Greenberger and Lowe describe two cases of patients satisfied with sensation, cosmesis, and function in 1999 but do not provide a method of quantifying their utility [54]. Further investigation into patient satisfaction with this procedure would be advantageous for patient counseling.

#### 3.2.4. Perineal Urethrostomy

For patients receiving total penectomy for penile cancer, perineal urethrostomy provides adequate urinary diversion. Data gathered from 246 patients internationally who underwent this reconstruction by Slongo et al. highlight both the effectiveness of the diversion and its potential complications [55]. They found that 65% of the patients were free of postoperative complications, and of those who did experience problems, wound infection (15%) was most prevalent, followed by dehiscence (5%) and tissue necrosis (2%). A total of 35 of the 246 (14%) patients experienced perineal urethrostomy stenosis (PUS), unexplained by the tumor staging. However, they did find a correlation between adjuvant chemo and radiation therapies and the incidence of PUS, which may relate to the disease severity of this subset of the study population as well as to poor wound healing. The PUS was treated with urethral dilation, and corrective surgery was also indicated in a majority of these patients [55]. de Vries et al. also conducted an international retrospective study including 299 patients, in which 19% experienced complications postoperatively, with wound infections and dehiscence being most prevalent. Moreover, 12% of this entire patient population experienced PUS, 74% of whom then underwent revision [56].

Falcone et al. analyzed the outcomes of perineal urethrostomy retrospectively in ten patients from a single center. Here, 20% endured postoperative complications, and a single patient experienced recurrence [57]. Overall, perineal urethrostomy is a relatively safe and effective reconstructive avenue for patients post-penectomy, assuming they are adequately counseled regarding the risk of complications.

#### 3.2.5. Phalloplasty and Penile Implant

Total phallic reconstruction (TPR) and penile implantation are other options for patients who have undergone radical penectomy, ideally allowing for more normal erogenous and tactile stimulation experiences of micturition and sexual intercourse [58]. Phalloplasty was pioneered by Bogoras in 1936, using abdominal flaps and rib cartilage to create a penis without a urethra [59,60]. Many advances have been made since to reach the concept of an integrated urethra, which was first implemented by Chang and Hwang in 1972 [61]. They created a urethra by tubularizing the proximal thigh and lower abdominal skin around a catheter. The procedure now most commonly involves the creation of a urethra from a radial artery free flap (RAFF) [60]. The ulnar artery is also used, although less commonly. Aside from an abdominal or radial artery flap, another option is an anterolateral thigh flap, ideal for men with scant subcutaneous fat due to its prevalence in that area [58]. A penile implant is sometimes indicated following phalloplasty for patients desiring improved sexual function outcomes [62].

A retrospective study by Garaffa et al. analyzed the results of fifteen patients who received RAFF TPR [60]. All but one were subsequently able to urinate standing, and each of the fifteen were cosmetically pleased. Five of seven who received a prosthesis were able to participate in penetrative intercourse. The most common complications arising from these procedures are urethral strictures and fistulae [58,60].

Young et al. studied the outcomes of patients who underwent RAFF phalloplasty, with and without subsequent penile implant [62]. It is important to note that these procedures were indicated for penile deficiency and not penile carcinoma, and hence the median age was much lower (19.7 years old for RAAF with implant, 23.5 years old for RAFF only). Nine patients who underwent RAFF and penile prosthesis implantation completed the researchers’ surveys, and they received responses from four patients who underwent only RAFF. While the interpretation of the results is limited by the small sample size, they found no apparent difference in the scores for penile perception or sexual quality of life between the patients who received penile implantation and those who did not. They did report an increase in overall sexual functioning, as determined by the IIEF questionnaire [62]. Overall, phalloplasty with or without penile implants can prove beneficial to quality of life for men with penile cancer post-penectomy (Table 2).

## 4. Methodological Challenges and Strengths

The challenges of this review are tied to the inherent weaknesses of the examined evidence. Definitive conclusions cannot be drawn due to the heterogeneity of the evidence presented herein. Multiple studies include patients of varying clinical and pathologic stages, making the subgroups small and comparison of the data difficult. Reconstructive techniques are presented more descriptively than scientifically, and controlled trials comparing techniques and their outcomes and complications are lacking in the literature. Additionally, the retrospective nature of the majority of the included evidence regarding local therapy for penile SCC is inferior to the minority of randomized controlled trials in the space.

## 5. Conclusions

In conclusion, penile SCC represents a rare malignancy with acceptable local treatment options and robust CSS rates when detected early. The options for local treatment vary based on the clinical and pathologic staging, and they range from topical antineoplastics to partial and radical penectomy. Several options offer preservation of upright voiding and sexual function, including topical chemotherapy, laser therapy, Mohs surgery, glansectomy, wide local excision, and partial penectomy. When utilized in an appropriate clinical context, the modalities offer excellent survival outcomes. Timely diagnosis and access to treatment are crucial for addressing the disease effectively. The more radical therapies, including glansectomy, extensive wide local excision, and partial/radical penectomy, demand practical and functional reconstructive techniques. This role is largely filled by some combination of skin grafting, creation of a neo-glans, creation of a functional penile stump, perineal urethrostomy, and phalloplasty, the last of which can be combined with penile implant for the preservation of sexual function in otherwise impotent patients.

## Figures and Tables

**Figure 1 cancers-16-02704-f001:**
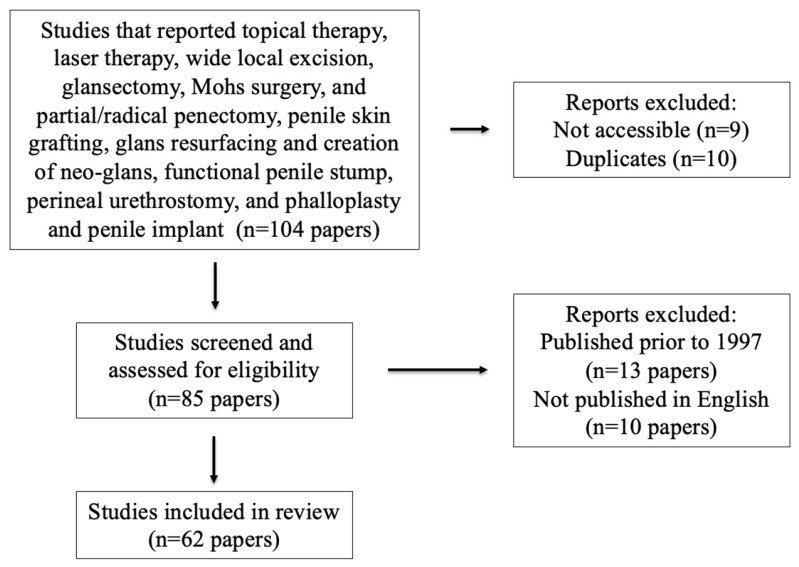
Summary of the literature search.

**Table 1 cancers-16-02704-t001:** Summary of the published data describing local therapy for penile cancer.

Author, Year	Treatment Modality	Results
Hajiran et al., 2021 [12]	5-FU	Complete response in 65%, partial response in 25%, and no response in 10% at 18-month median follow-up; median recurrence-free survival of 14 months.
Manjunath et al., 2017 [14]	5-FU and Imiquimod	10-year 50% complete response rate for 5-FU and 44% for Imiquimod; 31% partial response for 5-FU.
Torelli et al., 2017 [16]	Imiquimod then CO_2_ laser	Ten patients followed for mean of 26 months; no residual tumor in six, stable disease in two, and progressive disease in two.
Bandieramonte et al., 2008 [17]	CO_2_ laser	10-year recurrence rate of 17.5% with amputation rate of 5.5%.
Meijer et al., 2007 [21]	Laser therapy	44 patients followed for mean of 52.1 months; local recurrence in 48%, 20% recurrence elsewhere on the glans penis; nodal metastases in 10 patients, 8 of whom had T2 disease on biopsy.
Windahl et al., 2004 [22]	Combination CO_2_ and Nd:YAG laser	67 men with median follow-up of 42 months; during follow-up, 59 alive and 8 had died of penile carcinoma (2) and concurrent disease (6). Of 13 patients (19%) with local recurrence, 10 underwent repeat laser treatment successfully.
Sosnowski et al., 2016 [27]	Wide local excision	pT1 and T2: local recurrence is relatively common, but five-year RFS is 63%.
Baumgarten et al., 2018 [28]	Glansectomy	282 pT2 patients in cohort, 60.9% were treated with glansectomy; 85.7% of the pT2 patients recurred 5 years.
Sakalis et al., 2022 [30]	Glansectomy	Review of 1681 men (86.4% pT1-2); five-year RFS from 78.0–95.8%.
Alcala et al., 2022 [33]	Mohs surgery	43 patients with Ta-T2 disease (including Tis); overall local recurrence rate was 2% (n = 1) at median 47 months; local recurrence in none with Ta-T1 disease at 5 years; local recurrence for T2 patients 14% at one year.
Veeratterapillay et al., 2015 [35]	Partial/radical penectomy	Median follow-up of 61 months; penile-preserving surgery (49%), partial penectomy (24%), radical penectomy (24%), and chemotherapy or radiotherapy for metastatic disease (3%); organ-preserving surgery: local recurrence rate 18% (4% for amputative surgery), with 94% of recurrences within three years; five-year CSS of 85% and a 10-year CSS of 81%.
Djajadiningrat et al., 2014 [36]	Local therapy/partial/radical penectomy	859 patients (pT1-3) treated from 1956 to 2012; penile amputations decreased; 5-year local recurrence after penile preservation was 27% (3.8% after partial penectomy). Preservation no different compared to (partial) amputation; penile preservation group → local recurrence was not associated with CSS.

**Table 2 cancers-16-02704-t002:** Summary of the published data describing reconstruction after penile SCC excision.

Author, Year	Treatment Modality	Results
Triana et al., 2017 [39]	Penile Skin Grafting	Both split- and full-thickness skin grafting are reasonable options for reconstruction after glansectomy, wide local excision, or Mohs surgery; higher graft failure rate in full-thickness grafts and graft must be 10–20% larger due to contracture.
Pappas et al., 2019; Palminteri et al., 2011; O’Kelly et al., 2017; Weibel et al., 2018 [45,46,49,50]	Glans Resurfacing and Creation of Neo-glans	Total glans resurfacing is a promising reconstructive option for localized, superficial penile cancer (CIS, Ta, and T1 grade 1/2); complication rate of <5% and graft-take rate of nearly 100%; neo-glans created with either split-thickness skin graft, acellular dermal matrix, or tunica vaginalis graft.
Korkes et al., 2010 [52]	Creation of Functional Penile Stump	Following amputation, use the “parachute” technique: ventral urethral spatulation and suturing the “V” skin flap from ventral to dorsal, so as to approximate the edges while preserving a urethral meatus; retention of the most penile function and universal approach.
Slongo et al., 2020 [55]	Perineal Urethrostomy	246 patients had perineal urethrostomy: 65% were free of postoperative complications (wound infection (15%), dehiscence (5%) and tissue necrosis (2%)). A total of 35 out of 246 (14%) patients experienced perineal urethrostomy stenosis (PUS).
De Vries et al., 2021 [56]	Perineal Urethrostomy	299 patients had perineal urethrostomy: 19% experienced complications, with wound infections and dehiscence being most prevalent; 12% experienced PUS, 74% of whom underwent revision.
Garaffa et al., 2009 [60]	Phalloplasty and Penile Implant	15 patients received radial artery free flap total phallic reconstruction; 14 voided standing; all were cosmetically pleased; five of seven who received a prosthesis were able to participate in penetrative intercourse; common complications are urethral strictures and fistulae.

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
