# Peer review of "Local Therapy and Reconstruction in Penile Cancer: A Review"

_cancers, 2024, doi:10.3390/cancers16152704_

Round 1

Reviewer 1 Report

Comments and Suggestions for Authors

This article reviews the studies on the main treatments for penile cancer, from local therapy to radical penectomy, and reconstructive techniques. Local therapy can provide reliable oncological responses for patients with penile cancer.

Overall, this literature review is generally a qualified manuscript.

Author Response

Comments 1: 

This article reviews the studies on the main treatments for penile cancer, from local therapy to radical penectomy, and reconstructive techniques. Local therapy can provide reliable oncological responses for patients with penile cancer.

Overall, this literature review is generally a qualified manuscript.

Response 1: Thank you for pointing this out. We agree with this comment. No changes have been made based on the comment. We appreciate your encouragement for our publication.

Reviewer 2 Report

Comments and Suggestions for Authors

I read with great interest this paper on Local Therapy and Reconstruction in Penile Cancer.

The review is well structured and overall fluent in the reading.

However, some revisions should be made in order to improve the quality of the paper. 

First of all a PRISMA diagram should be presented in the methods section.

Second: a brief paragraph on quality of life is missing

I also suggest adding the following paper that authors missed in their search on PubMed  doi: 10.3390/curroncol30120765. 

Check typos

Author Response

Comments 1: 

First of all a PRISMA diagram should be presented in the methods section.

Response 1: Thank you for pointing this out. We agree with this comment. Therefore, we have added a PRISMA diagram. This change can be found on line 85 in the manuscript.

Comments 2: 

Second: a brief paragraph on quality of life is missing

Response 2: Thank you for pointing this out. We agree with this comment. Therefore, we have added a citation in the introduction section. This change can be found on lines 45-49 in the manuscript. "In fact, some men who have undergone therapy for penile SCC retrospectively prioritize maintenance of potency over long-term survival as the effects of penectomy on quality of life are significant [8]. The aggressiveness of therapy correlates conversely with quality of life with regard to overall health status, sexual health, and physical functioning [9]."

Comments 3: 

I also suggest adding the following paper that authors missed in their search on PubMed  doi: 10.3390/curroncol30120765. 

Response 3: Thank you for pointing this out. We agree with this comment. Therefore, we have incorporated this manuscript. This change can be found on lines 190-192 in the manuscript. "Cilio et al. corroborate these results in 34 patients, with patients who underwent WLE versus glansectomy demonstrating a higher rate of erectile dysfunction and sexual impairment [26]."

Reviewer 3 Report

Comments and Suggestions for Authors

The author introduced in detail the local treatment methods and postoperative reconstruction methods of penile malignant tumors, which is very interesting, but there are some concerns.

1. Although the method of local treatment is introduced in detail in the manuscript, it does not give good suggestions on how to choose the treatment method, so the clinical guidance significance is limited.

2. Lymph node dissection does not seem to have been mentioned by the author. Should lymph node dissection be performed instead, and how? What is the author's opinion.

3. Does local treatment include local radiotherapy? There seems to be no discussion of this in the author's manuscript.

Author Response

Comments 1: 

1. Although the method of local treatment is introduced in detail in the manuscript, it does not give good suggestions on how to choose the treatment method, so the clinical guidance significance is limited.

Response 1: Thank you for pointing this out. We do not entirely agree with this comment. Each subsection reviews the clinical stage of disease that can be managed with each local therapy.  The manuscript language has been modified for clarification.

Comments 2: 

2. Lymph node dissection does not seem to have been mentioned by the author. Should lymph node dissection be performed instead, and how? What is the author's opinion.

Response 2: Thank you for pointing this out. We agree with this comment, although it does not pertain to local therapy.  Lymph node dissection should be performed prophylactically in patients with pT1b disease or higher or those with clinically evident nodes.  It should never be performed instead of local therapy.

Comments 3: 

3. Does local treatment include local radiotherapy? There seems to be no discussion of this in the author's manuscript.

Response 3: Thank you for pointing this out. We agree with this comment. Therefore, we have added a brief section at the conclusion of the local therapy section on radiotherapy. This change can be found on lines 250-262 in the manuscript. "Radiation has a limited role in local therapy for penile cancer, but is an option for invasive penile SCC (>=T1, grade 1-2). Both EBRT and brachytherapy can be utilized in the post-circumcision setting for T1-2, N0 disease. Concurrent chemotherapy is often administered in this setting, and should universally be used in T3-4 or N+ disease. Post-surgical EBRT is also an option with positive primary site margin following penectomy [10]. EBRT delivers radiation doses of 60-75 Gy with 5-year local control rates of 40-70%. CSS in reported series range from 58-86% [15]. Brachytherapy can be used to deliver radiation doses from 50-65 Gy. 5-year local control rates in reported series range from 70-87% and 5-year CSS of 72-88%. Well over half of patients are able to maintain their penises during follow up periods [15]. Important considerations in radiation patients are the above-described circumcision prior to therapy to avoid phimosis as well as late sequelae including meatal stenosis and ulceration of the penile skin and soft tissue [15]."

Round 2

Reviewer 2 Report

Comments and Suggestions for Authors

authors have addressed my major concerns. I endorse pubblication